# Emergence of Reassortment between a New and Reported Types of Betanodavirus in Shellfish

**DOI:** 10.3390/pathogens10101232

**Published:** 2021-09-24

**Authors:** Young Chul Kim, Joon Gyu Min, Kwang Il Kim, Hyun Do Jeong

**Affiliations:** 1Pathology Division, National Institute of Fisheries Science, Busan 46083, Korea; sai100@korea.kr; 2Department of Aquatic Life Medicine, Pukyong National University, Busan 48513, Korea; cdmin0621@hanmail.net (J.G.M.); kimki@pknu.ac.kr (K.I.K.)

**Keywords:** betanodavirus, reassortment, shellfish, KSNNV, type

## Abstract

Recently, three types of betanodavirus including red spotted grouper nervous necrosis virus (RGNNV), barfin flounder nervous necrosis virus (BFNNV), and Korean shellfish nervous necrosis virus (KSNNV) (proposed as a new fifth type) have been detected in shellfish in the marine environment around Korea. To investigate the presence of reassortment between betanodavirus types, the type based on the RNA2 segment of betanodaviruses carried in 420 domestic shellfish (*n* = 306) and finfish (*n* = 35), as well as imported shellfish (*n* = 79), was compared with the type identified by reverse-transcriptase polymerase chain reaction (RT-PCR) for RNA1 segment. Only five samples carrying reassortant betanodaviruses were found, appearing as RG/KSNNV (*n* = 2), KS/RGNNV (*n* = 1), and SJ/RGNNV (*n* = 2) types. From these samples, we successfully isolated two reassortant strains from Korean and Chinese shellfish in E-11 cells and called them KG1-reKS/RG and CM1-reRG/KS, respectively. In the full genome sequences, each RNA segment of the reassortant strains exhibited the same gene length and high sequence homology (≥98%) with the reference strains corresponding to the type of each segment. Both these reassortant strains induced high mortality to sevenband grouper (*Epinephelus septemfasciatus*) larvae with high viral concentrations in the body (10^9^ viral particles/mg) and severe vacuolation in the retina and brain. These are the first results showing the involvement of the KSNNV type in the reassortment of RNA segments in the reported types of betanodavirus, which could represent a new potential risk in fish.

## 1. Introduction

Betanodavirus is a causative pathogen of one of the most prevalent viral diseases in aquatic fish farmed around the world, causing viral encephalopathy, retinopathy (VER), and viral nervous necrosis (VNN). Over the past 30 years, a large number of fish species have been reported to be affected, especially in the larval stage, with necrosis of the brain and retina [1]. Betanodavirus is a small (24−34 nm in diameter), non-enveloped virus and contains a bi-segmented genome composed of two single-stranded positive-sense 3.1 kb RNA1 segments (encoding RNA-dependent RNA polymerase (RdRp) and protein B2 in subgenomic RNA3) and a 1.4 kb RNA2 segment (encoding coat protein (CP), which are capped but not polyadenylated [2,3,4]. Betanodaviruses have generally been classified into four major types by the International Committee on Taxonomy of Viruses (ICTV) as red-spotted grouper (RGNNV), barfin flounder (BFNNV), striped jack (SJNNV), and tiger puffer nervous necrosis virus (TPNNV) types [5]. Within the classification of betanodaviruses, SJ/RGNNV (RNA1 segment: SJNNV; RNA2 segment: RGNNV) and RG/SJNNV (RNA1 segment: RGNNV; RNA2 segment: SJNNV), genetically recombined between the RNA segments of RGNNV and SJNNV types as the reassortment, have also been reported to cause serious damage to the fish farming industry located on the Mediterranean coast [6,7,8,9].

A high frequency of BFNNV as well as RGNNV types has also been reported in various domestic and imported shellfish, which is different from the previous reports of only the RGNNV type in cultured finfish in Korea. Moreover, the wide distribution of a new betanodavirus, Korean shellfish nervous necrosis virus (KSNNV), which was suggested to belong to a new fifth type, was reported through surveillance studies using shellfish species as viral reservoirs in the marine environments of Korea and China [10,11]. Thus, at least three betanodavirus types (RGNNV, BFNNV, and KSNNV) exist in the marine environment of East Asia, so it may strongly implicate the generation of new and diverse reassortant viruses by mutual genetic reassortment.

However, so far, there is a lack of information on the presence of reassortment between betanodavirus types in the aquatic environment in Asia because most of the studies on betanodavirus surveillance have mainly been conducted to detect betanodavirus based on the nucleotide sequence of the RNA2 segment only [12,13]. Recently, various betanodavirus types including KSNNV type were identified in the marine environment in Asia. Thus, it is important to investigate the emergence of the new reassortment between betanodavirus types and their risk to aquaculture. We established a new detection and genotyping system based on types of RNA1 and RNA2 segments simultaneously and examined the pivotal role of the KSNNV type in the emergence of reassortment between types of betanodavirus in the marine environment.

## 2. Results

### 2.1. Identification of Reassortment between the Types of Betanodavirus in Shellfish

In our laboratory, 420 samples were previously collected using two consecutive RT-PCRs, RNA2-detection semi-nested RT-PCR (R2-DSN-2 RT-PCR), and RNA2-discriminative multiplex RT-PCR (R2-DMT RT-PCR) for the detection and discrimination of betanodaviruses in shellfish based on the nucleotide sequence of the RNA2 segment. At this time, the amplicon size used for type identification in R2-DMT RT-PCR was 211, 292, 347, 413, and 468 for KSNNV, BFNNV, SJNNV, TPNNV, and RGNNV, respectively [10,11]. If the virus showed the reassortment, the type for the RNA2 segment identified by R2-DMT RT-PCR in previous experiments is different from the type determined by the RNA1-discriminative multiplex RT-PCR (R1-DMT RT-PCR) developed and used in this study. Such type mismatches between RNA1 and RNA2 segments were found in four domestic and one imported Chinese shellfish (Table 1). The nucleotide sequences determined for each RNA1 and RNA2 segment of these five samples and phylogenetic analysis revealed the presence of genetic reassortment: SJ/RGNNV (RNA1 segment: SJNNV; RNA2 segment: RGNNV) (from two domestic common orient clam *Meretrix meretrix*), KS/RGNNV (RNA1 segment: KSNNV; RNA2 segment: RGNNV) (from a one domestic granular ark *Tegillarca granosa*), and RG/KSNNV (RNA1 segment: RGNNV; RNA2 segment: KSNNV) (from a one domestic Pacific oyster *Crassostrea gigas* and a one imported Manila clam *Ruditapes philippinarum* from China) (Table 1). Fortunately, there was no identified reassortment in 35 positive betanodaviruses detected in domestic finfish. As a result, the detection rates of reassortant virus from shellfish were 0.5% (4/741) in Korea and 0.7% (1/138) in China, but none were detected in Japan (0/90). All nucleotide sequences of these reassortant betanodaviruses were deposited to the National Center for Biotechnology Information (NCBI) GenBank (accession no. MG366159, MG366160, KX601151, KX601152, MG387121, MG387122, MG387123, and MG387124).

### 2.2. Isolation of Reassortant Viruses in Shellfish

Double-blind passages on E-11 were performed with the digestive tracts of five shellfish samples (one Pacific oyster, two common orient clam, one granular ark, and one Manila clam) showing a type mismatch between the RNA1 and RNA2 segments in type-discrimination RT-PCRs. Only two samples, the granular ark in Korea and the Manila clam imported from China, produced sufficient CPE to indicate successful viral isolation under in vitro conditions and were named as the KG1-reKS/RG and CM1-reRG/KS strains, respectively. A specific band was also found in each type-discriminating multiplex RT-PCR on supernatants of cultured viruses. The supernatant of KG1-reKS/RG produced an 843 bp band for KSNNV in R1-DMT RT-PCR and 468 bp for RGNN in R2-DMT RT-PCR, and the supernatant of CM1-reRG/KS also produced a 214 bp band for RGNNV in R1-DMT RT-PCR and 211 bp for KSNNV in R2-DMT RT-PCR. Thus, we successfully obtained two reassortant viruses as pure isolates, one present in domestic shellfish as the KS/RGNNV type and the other present in imported shellfish as the RG/KSNNV type. The CPE form was identical to the typical CPE generated by RGNNV and KSNNV types (Figure 1). In addition, the negative staining of the virion in the carbon-coated formvar of isolated KG1-reKS/RG and CM1-reRG/KS strains showed a viral particle of typical betanodavirus shape with a 30−35 nm, unenveloped spherical form (Figure 2).

### 2.3. Comparision of the Full-Sequence Homology with Other Betanodaviruses

The total length, open reading frame (ORF; RdRp and CP genes), and untranslated region (UTR) of each segment of KG1-reKS/RG (RNA1 segment 3123 bp; RNA2 segment 1434 bp) and CM1-reRG/KS (RNA1 segment 3105 bp; RNA2 segment 1444 bp) strains were the same as the those of KSNNV-KOR1 RNA1 segment (3123 bp)/SGWak97 RNA2 segment (1434 bp) and SGWak97 RNA1 segment (3105 bp)/KSNNV-KOR1 RNA2 segment (1434 bp), respectively (Table 2). The homology of the full nucleotide sequence of the RNA1 (sub-genomic RNA3) segment of KG1-reKS/RG and CM1-reRG/KS strains were 99.8% (99.5%) and 97.2% (94.1%) identical to that of the corresponding segments in KSNNV-KOR1 and SGWak97, respectively. In the RNA2 segment, the homology was determined to be 97.4% and 98.5% with the corresponding segments in SGWak97 and KSNNV-KOR1, respectively (Table 3). The high nucleotide sequence homology with the corresponding RNA segments of the representative viruses of betanodavirus was also seen in the tendency in the alignment of the deduced amino acid sequence (Table 3). In contrast, the nucleotide sequence homology of both the RNA1 and RNA2 segments with other types (BFNNV, SJNNV, and TPNNV) was less than 89% (Table 3). In the phylogenetic analysis, each segment of both reassortant strains, the RNA1 and RNA2 segments of KG1-reKS/RG and RNA2 and RNA1 segments of CM1-reRG/KS, was also clustered with KSNNV-KOR1 and SGWak97 and was completely distinguished from the other types of betanodavirus (Figure 3). Thus, mismatches in type showing high nucleotide and amino acid homology between each of the two RNA segments suggested the emergence of new reassortment between the KSNNV and the RGNNV types.

### 2.4. Pathogenicity of Reassortant Viruses against Sevenband Grouper Larvae

From 2–3 days post-inoculation (dpi), abnormal swimming of larvae appeared in all temperature groups. Larvae infected with either the KG1-reKS/RG or CM1-reRG/KS strain showed 100% cumulative mortality five days after the challenge at different temperatures (20 and 25 °C), indicating marked pathogenicity of 24% and 29%, respectively, compared to the mock groups (Figure 4a). All pooled samples from virus-inoculated larvae showed betanodavirus-positive results in qRT-PCR. No specific amplicon in qRT-PCR was generated from pooling samples in the mock groups as the negative control (data not shown). The peak viral concentrations in the body of the infected larvae 4–5 days after inoculation ranged from 8.9 × 10^8^ to 1.1 × 10^9^ viral particles/mg in the challenge groups (Figure 4b). Although the peak viral particles of the infected larvae based on reassortant viruses or water temperatures showed no significant difference between the challenge groups, the viral concentrations tended to increase at 25 °C compared to 20 °C. With high mortality in both reassortant virus-infected larvae, the moribund fish at 3 dpi showed typical histopathological traits of betanodavirus infection including vacuolation in the retina and brain (Figure 5.). There were no histopathologic abnormalities at 5 dpi in the negative control group.

## 3. Discussion

In the last three decades since VER was reported for the first time in 1989, only the RGNNV type has been detected in symptomatically infected fish in Korea [1,14,15]. Recently, however, because of filter-feeding activity and the use of shellfish species as bioindicators, various pathogenic agents have been identified [10,11,13,16], and a new betanodavirus (KSNNV type) with a wide distribution in marine environments was isolated [17]. Additionally, several laboratories in Europe [8,18,19] reported the emergence of reassortment harboring different genotypic segments in RNA1 and RNA2 of betanodavirus. Most of these reassortant viruses were related to only the RGNNV and SJNNV types. However, when we consider that the KSNNV type might be a common ancestor of betanodaviruses with a wide range of distribution in aquatic environments [16], this betanodavirus type, separated from the four known major types, is highly likely to play an important role in the emergence of new reassortment by making genetic exchanges with the gene segments of other betanodaviruses easier. Thus, to investigate reassortment between types of betanodavirus identified from domestic and imported shellfish samples, we compared segment types using two type-discrimination multiplex RT-PCR, one for the RNA1 (R1-DMT RT-PCR) and the other one for the RNA2 segment (already determined by Kim et al. [10]) and obtained five samples showing intersegmental genotypic mismatches among 420 marine samples (Table 1). The reassortant viruses detected in domestic and imported Chinese shellfishes indicated a strong possibility of genetic reassortment between various types of betanodavirus, including a new KSNNV type in the marine environment of the world including China. Fortunately, there was no identified reassortment between betanodaviruses detected in domestic finfish.

In the five samples carrying candidate reassortant viruses, we successfully isolated two new infectious reassortant strains under in vitro cultural conditions (Figure 1), showing 32–35 nm typical betanodavirus shapes by electron microscopy (Figure 2), a KG1-reKS/RG (KS/RGNNV type) strain in the domestic granular ark, and a CM1-reRG/KS (RG/KSNNV type) strain in the Manila clam imported from China. To our knowledge, it was the first report showing the emergence of new reassortment between new and existing types of betanodaviruses in the marine environment. Moreover, although cultural isolation was not successful in this study, the detection of the SJ/RGNNV type as the reassortant virus in domestic mussels was a surprising result because there have been no outbreak reports of the SJNNV type in farmed finfish in Korea.

The results of the full-genome sequences determined for each segment of KG1-reKS/RG and CM1-reRG/KS strains by RACE showed very high homology with the corresponding segment of each corresponding reference type. In particular, the sequence homology of the KSNNV RNA1 or RNA2 segments in the two reassortant strains compared with strains of other KSNNV type subfamilies showed much higher homology than the homology between the other corresponding segment types (RGNNV type) (Table 3). It is possible to infer that there may be fewer genetic changes in each segment of the KSNNV type, which can be said to be a kind of ancient betanodavirus type, than in the RGNNV type.

To understand the transmission risk of reassortant virus isolated from shellfish, we tried to determine their pathogenicity via the immersion method with farmed fish, especially sevenband grouper (*Epinephelus septemfasciatus*), known as one of the hosts susceptible to betanodavirus [20]. No significant differences were detected in the cumulative mortality or viral growth patterns in sevenband grouper larvae infected with two reassortant strains (KG1-reKS/RG and CM1-reRG/KS) at temperatures of 20 and 25 °C (Figure 4). The infected larvae also showed typical histological changes in the retina and brain as VNN infections (Figure 5). Although the RGNNV (SGWak97 strain derived from sevenband grouper) type could replicate at 20 °C in clones derived from SSN-1 cells, the optimal growth temperature for that type ranged from 25 to 30 °C [21]. Meanwhile, the same type of virus (GNNV strain derived from grouper *Epinephelus coioides*) did not produce CPE at 20 °C in GF-1 cells [22,23], indicating that temperature impacted viral replication by RdRp activity [24]. Our previous study showed that two betanodaviruses derived from shellfish (KSNNV type, KSNNV-KOR1 strain; RGNNV type, SFRG10/2012BGGa1 strain) could also replicate in E-11 cells and showed high mortality of sevenband grouper larvae at both 20 and 25 °C [17]. In this study, even in the challenge groups at 20 °C, which is not the optimal growth temperature for RGNNV, two different reassortant strains (KS/RGNNV or RGNNV/KSNNV type in RNA1 and RNA2 segments) were fully replicated in E-11 cells, leading to the high mortality of sevenband grouper larvae, consistent with the pathogenicity of single-type virus infection described in a previous study [17]. Betanodavirus mainly led to clinical infections in susceptible fish at the larval and juvenile stages [24,25]. Although we determined the pathogenicity in sevenband grouper larvae by the combined genetic KSNNV and/or RGNNV type in RNA1 and/or RNA2 segments, it is presumed that genetic reassortment did not affect viral adsorption and replication in either E-11 cells or sevenband grouper larvae. Thus, these reassortant betanodaviruses are pathogens that can cause an emerging disease that could cause great damage to the hatchery industry of sevenband grouper.

Viruses having multiple genetic segments, such as the influenza virus, can generate new various reassortant viruses that could cause emergent diseases with a high probability of super-infection in the host species [26]. It is difficult to predict the emergence of such reassortant virus in the marine environment before an outbreak in aquaculture, but it is very important to establish a predictable surveillance tool in advance. Of note, to prevent industrial damage by the emergence of reassortment between betanodaviruses in aquaculture, enhanced surveillance based on type targeting in fish, as well as shellfish, should be performed. Additionally, risk assessment of reassortant virus based on the fish cultivation stages including juvenile and adult susceptible hosts is necessary. In this study, a new genetic reassortment between betanodaviruses was identified from shellfish in the marine environment in which three types (RGNNV, BFNNV, and KSNNV) existed simultaneously, as hypothesized, from contact between different types of betanodavirus, which could induce new reassortment. From this point of view, the use of shellfish is suggested as a bioindicator to effectively predict emerging diseases and assess their risk in aquaculture in advance.

## 4. Materials and Methods

### 4.1. Betanodaviruses in Shellfish and Finfishes

In a previous study [10,11], betanodaviruses were identified from shellfish (domestic shellfish from 2011 to 2014; Chinese and Japanese shellfish imported from 2012 to 2015) and domestic farmed finfishes (from 2011 to 2013) based on the RNA2 segment. Briefly, betanodaviruses were detected using R2-DSN-2 RT-PCR with primers (c2VNNF1, c2VNNF2 and c2VNNR) (Table 4). Then, the type was determined by R2-DMT RT-PCR with primers targeting different betanodavirus types (mixture of c2VNNF2 as a common-sense primer for all types, and 5 different antisense primers: s2VNNKSR for KSNNV, s2VNNRGR for RGNNV, s2VNNBFR for BFNNV, s2VNNSJR for SJNNV, and s2VNNTPR for TPNNV). Of these, a total of 420 samples (306 domestic shellfish (179 Pacific oyster *Crassostrea gigas*, 49 mussel *Mytilus edulis*, 29 Manila clam *Ruditapes philippinarum*, 28 common orient clam *Meretrix meretrix,* and 21 granular ark *Tegillarca granosa*); 35 domestic finfishes (23 olive flounder *Paralichthys olivaceus*, 3 rock bream *Oplegnathus fasciatus*, 2 red sea bream *Pagrus major*, 2 rockfish *Sebastes schlegeli,* and 5 stone flounder *Platichthys stellatus*); 48 Chinese shellfish (14 Manila clam, 13 common orient clam, 13 adams venus clam *Mercenaria mercenaria*, 3 venus clam *Mercenaria stimpsoni*, 3 wrinkled venus clam *Callista brevisiphonata*, 1 scallop *Saxidomus purpurata,* and 1 Chinese cyclina *Cyclina sinensis*); 31 Japanese shellfish (5 Manila clam, 5 venus clam, 9 granular ark, and 12 scallops)) (Table 1) were found to contain only a single type using R2-DMT RT-PCR and were stored at –80 °C until they were used for RT-PCRs based on the RNA1 segment instead of the RNA2 segment.

### 4.2. Primer Design for Genotyping of Betanodaviruses Based on RNA1 Segment

To identify and discriminate the type based on the RNA1 segment of betanodavirus, we developed two consecutive RT-PCRs: semi-nested two-step RT-PCR (R1-DSN-2 RT-PCR) for the detection of the RNA1 segment in all different types of betanodavirus and then R1-DMT RT using the amplicons from R1-DSN-2 RT-PCR as a template for genotyping of RNA1 segments based on the length of the generated fragments. Specifically, for the R1-DSN-2 RT-PCR as the semi-nested RT-PCR, the first-round PCR amplification was performed using c1VNNF1/c1VNNR primers targeting the conserved region in the RNA1 segment of the five different betanodavirus types, followed by the second-round PCR amplification, which was performed using c1VNNF2/c1VNNR primers (Table 4). To determine the type of betanodavirus, the amplicons from the first round of R1-DSN-2 RT-PCR were used as templates for R1-DMT RT-PCR. With a common-sense primer (c1VNNF2), antisense primers (s1VNNKSR, s1VNNRGR, s1VNNBFR, s1VNNSJR, and s1VNNTPR) designed from the type-specific region amplified in the first round of R1-DSN-2 RT-PCR were used for R1-DMT RT-PCR. The type was determined as the size of amplicon generated in R1-DMT RT-PCR, and the expected size for each type was 843, 214, 448, 361, and 288 bp for KSNNV, RGNNV, BFNNV, SJNNV, and TPNNV, respectively. All primers were designed using BioEdit software (ver. 7.0.6 Department of Microbiology, North Carolina State University, Raleigh, NC, USA) and Primer3 software (http://primer3.ut.ee/) and were synthesized with an automated DNA synthesizer (Bioneer, Daejeon, Korea) by the phosphoramidite method.

### 4.3. RT-PCR

Total RNA was extracted from 25 mg of shellfish digestive tissues and 100 µL of the virus isolate cell supernatants with the RNeasy Plus Mini Kit (Qiagen, Germantown, MD, USA), according to the manufacturer’s instructions. For cDNA synthesis, isolated total RNA (1 µg) was heated at 70 °C for 5 min, followed by the addition of Moloney murine leukemia virus reverse transcriptase (M-MLV, Promega, Madison, WI, USA) with 1.25 mM random primers (Promega, Madison, WI, USA) at 42 °C for 60 min. The reacted sample was heated at 95 °C for 5 min to inactivate the reverse transcriptase. RT-PCR was performed in a 50 µL reaction mixture containing 2 µL of obtained cDNA, 5 µL of 10× PCR buffer (1.5 mM MgCl_2_), 200 μM of each dNTP, 1 µM of each primer set (Table 4), 1.25 U AmpliTaq DNA polymerase (Perkin-Elmer, Norwalk, CT, USA), and diethylpyrocarbonate (DEPC)-treated water added to achieve the final volume, and analyzed using a Perkin-Elmer 2400 thermal cycler (Perkin-Elmer, Norwalk, CT, USA). The c1VNNF1/c1VNNR and c1VNNF2/c1VNNR sets were used to perform R1-DSN-2 RT-PCR for the first round and second round of RT-PCR, respectively. For R1-DMT RT-PCR, c1VNNF2 and the five antisense primers were used at once (Table 4). The PCR conditions for the first and second round included a pre-denaturation at 95 °C for 5 min, 35 cycles at 95 °C for 30 s, 55 °C for 30 s, and 72 °C for 30 s, followed by an extension period at 72 °C for 7 min. The amplified PCR products were subjected to 2% agarose gel electrophoresis.

### 4.4. Quantitative RT-PCR

Nucleic acid from the brain tissue of the experimental larval fish (pooled 10 larvae) was extracted with the RNeasy Plus Mini Kit (QIAGEN, Hilden, Germany) and cDNA synthesis was performed according to the manufacturer’s instructions. The concentration of viral particles in the experimental fish was determined by quantitative (q)RT-PCR using the LightCycler 480II instrument (Roche, Mannheim, Germany). The qRT-PCR reaction mixture contained 1 µL of cDNA, c2VNNF2, and q2VNNR primers at a concentration of 500 nM each, and the LightCycler 480 SYBR Green Master mixture (Roche, Mannheim, Germany) (Table 4). The qRT-PCR conditions were as follows: 95 °C for 10 min, followed by 40 cycles of 95 °C for 10 s, 60 °C for 15 s, and 72 °C for 20 s. As a positive control, recombinant plasmids containing 126 bp from the RNA2 segment of betanodavirus (amplified using the c2VNNF2/q2VNNR primer set) were cloned from the transformed *Escherichia coli* DH5α. Serial 10-fold dilutions of the control plasmids were used to construct a standard curve (1 × 10^8^ to 1×10^2^ copies/µL). The standard curves were calculated using the mean data from tests performed in triplicate and indicated a good linear relationship between the Ct values. All of the samples used were tested in duplicate.

### 4.5. Determination of Full-Genomic Sequences of Reassortant Virus

To determine the partial sequence of each RNA segment in the reassortant virus, PCR amplicons generated from R1- and R2-DSN-2 RT-PCRs were purified by 2% agarose gel electrophoresis with the Expin^TM^ Gel SV Kit (GeneAll, Seoul, Korea) and cloned into the TOPO-TA vector (Invitrogen, Carlsbad, CA, USA) according to the manufacturer’s protocol. Each recombinant plasmid was sequenced using the Big Dye Terminator Cycle DNA Sequencing Kit (ABI PRISM, Applied Biosystems, Foster City, CA, USA) and an automatic sequencer (ABI3730XL DNA analyzer, Applied Biosystems, Foster City, CA, USA). To identify the complete sequence of the viral segments, the 5′ and 3′ termini of the amplicons generated from the RNA1 and RNA2 segments in two types of reassortant virus were explored by rapid amplification of the cDNA ends (RACE) using a GeneRacer^TM^ Kit (Invitrogen, Carlsbad, CA, USA) with the designed primers (Table 4) according to the manufacturer’s instructions. For 3′ RACE, a poly(A) tail was added to the extracted viral RNA using a Poly(A) Tailing Kit (Ambion, Austin, TX, USA). For both ends, cDNA was synthesized using Invitrogen^TM^ AMV Reverse Transcriptase (Invitrogen, Madison, WI, USA). The cDNA was then subjected to PCR with AmpliTaq DNA polymerase (Perkin-Elmer, Norwalk, CT, USA) using the gene-specific primers (GSP) designed from the partial nucleotide sequences of the amplicon determined from each segment (Table 4). The amplified PCR products of the 5′ and 3′ ends of the virus from RACE (RNA1 and RNA2) were purified from the agarose gel, cloned using a TOPO-TA vector according to the manufacturer’s protocol (Invitrogen, Carlsbad, CA, USA), transformed into chemically competent *E. coli* DH5α, and sequenced.

### 4.6. Comparison of Full-Genomic Sequences and Construction of Phylogenetic Tree

The complete sequences of the reassortant strains, KG1-reKS/RG and CM1-reRG/KS, were compared using BioEdit software (ver. 7.0.6) to determine the homology with the RNA1/RNA2 segment of five representative types of betanodavirus including KSNNV (MF170961/KT781097 of KSNNV-KOR1 from Pacific oyster, *Crassostrea gigas*) [11], RGNNV (AY324869/AY324870 of SGWak97 from sevenband grouper, *Epinephelus septemfasciatus*) [27], BFNNV (EU236146/EU236147 of JFIwa98 from barfin flounder, *Verasper moseri*), SJNNV (AB056571/AB056572 of SJNag93 from striped jack *Pseudocaranx dentex*), and TPNNV (EU236148/EU236149 of TPKag93 from tiger puffer, *Takifugu rubripes*) [27]. A phylogenetic tree was constructed by neighbor-joining analysis with MEGA X software (Center for Evolutionary Functional Genomics, The Biodesign Institute, AZ, USA) based on a bootstrap analysis with 1000 replicates. The complete sequences of various RNA1 and RNA2 segments from the betanodaviruses obtained from the GenBank database (http://www.ncbi.nlm.nih.gov/GenBank) were used to construct a phylogenetic tree.

### 4.7. Isolation of Reassortant Virus in Shellfish

To isolate the reassortant virus, the double-blind passage method was used [10]. Briefly, the digestive tracts of betanodavirus positive shellfish carrying the reassortant virus were added to 10 volumes (*w/v*) of phosphate-buffered saline (PBS, pH 7.4) and homogenized with a mini-homogenizer. After centrifugation (3000× *g* at 4 °C for 15 min), the supernatants were filtered through a 0.22 µm syringe filter (Advantec, Chiyoda city, Tokyo, Japan) and then diluted 10-fold with cold Hank’s balanced salt solution (HBSS, Sigma-Aldrich, St. Louis, MO, USA) to minimize the effect of cytotoxic components in the shellfish tissues. Aliquots (0.5 to 1.0 mL) of homogenates were inoculated onto drained 80% confluent E-11 (clone of the cell line SSN-1 (striped snakehead *Channa striatus* fry cell), ECACC 01110916) monolayers in 6-well plates and allowed to adsorb for 90 min at 20 °C. Plates prepared in triplicate for each sample were incubated at 15, 20, and 25 °C after the addition of L-15 medium supplemented with 10% fetal bovine serum (FBS), L-glutamine (2 mM), and antibiotics (100 IU mL^−1^ penicillin, 100 μg mL^−1^ streptomycin, and 0.25 μg mL^−1^ amphotericin B). With the observation of the cytopathic effect (CPE) by light microscopy at 400× magnification (Nikon, Tokyo, Japan), each culture was blind-passaged again after 10 days at culturing conditions identical to those described above. Seven days after the second round of double-blind passage, supernatants (0.5 mL) from the wells of the plate showing CPE were inoculated into fresh E-11 cells in T-25 flasks (Corning Glass Works, Corning, NY, USA) for virus isolation.

### 4.8. Transmission Electron Microscopy (TEM)

To clear the E-11 cell debris, the viral cultured supernatant was centrifuged at 10,000× *g* for 30 min and filtered using a 0.22 µm syringe filter (Advantec, Chiyoda city, Tokyo, Japan). The filtered supernatant was layered onto 5 mL of a 20% (*w/v*) sucrose-Tris-EDTA (TE) buffer cushion and centrifuged for 2 h in a swinging bucket rotor SW41-Ti rotor at 288,000 g using a Beckman Coulter OptimaTM L-100× P Ultracentrifuge (Beckman Coulter, Inc., Krefeld, Germany). The pellet was resuspended in 1 mL of 10 mM Tris buffer (pH 8.0) at 4 °C overnight. For electron microscopy, the resuspended viruses were stained with 1% uranyl acetate on formvar-coated copper grids (Sigma-Aldrich, St. Louis, MO, USA). Micrographs were taken with transmission electron microscopy (TEM, Hitachi H-7500, Tokyo, Japan) operating at 80 kV.

### 4.9. Pathogenicity of Two Reassortant Viruses to Fish in the Larval Stage

For the experimental challenge test of the sevenband grouper at the larval stage, fertilized sevenband grouper eggs (150 g) were purchased from a fish hatchery farm and confirmed to be betanodavirus-free by R1-DSN-2 RT-PCR. The eggs were rinsed five times using 25 °C heat-sterilized seawater and hatched under weak aeration at 25 °C in 300 L seawater in a 500 L tank. Seven days after hatching, visually healthy larvae were moved into small scoops from the hatching tank and counted to prepare 10 groups of larvae (500 larvae/group) using 10 mL pipettes (Corning, NY, USA). First, to analyze the cumulative mortality caused by two reassortant strains (KG1-reKS/RG and CM1-reRG/KS), a total of six experimental groups (three groups at two different temperatures, 20 and 25 °C) including mock groups were acclimatized for 24 h at their respective temperature of 20 or 25 °C. For the challenge test, viral infections were performed by the immersion method in each transparent aqua tank (5 L) at their respective temperature. Briefly, larvae were exposed to 2 L of heat-sterilized seawater containing each reassortant strain corresponding to 10^7^ viral particles/mL for 4 h at their respective temperature and then were transferred to 4 L of fresh seawater in each aqua tank (four groups). As negative control groups (two groups), untreated larvae were also kept in aqua tanks at their respective temperature. After immersion for 4 h, each group of larvae collected from a scoop was carefully washed in fresh heat-sterilized seawater for 2 h. During the experiment, feeding rotifers (10,000 cells/L) were supplemented, followed by the exchange of 30% of rearing water and the removal of feces at the bottom every 24 h. To observe the mortality in the experimental groups, the dead larvae sunk to the bottom were collected daily using transparent disposable 10 mL pipettes (Corning,) and counted to determine the number of dead larvae using a dissecting microscope (10×; Olympus, Tokyo, Japan). Second, to analyze the kinetics of viral replication and histological traits in viral-inoculated larvae, larvae (a total of four groups) were infected by the two reassortant strains using the immersion method as described above, followed by periodic sampling at 3, 5, 6, 10, and 14 dpi. In the periodic sampling groups, 10 larvae swimming in the rearing water were randomly picked up using a disposable 10 mL pipette under a shining light to determine the viral replication in the larval body. In all of the experimental groups, each experimental group was separated again into three groups for concurrent triplicate experiments. For histopathological analysis, samples fixed in 10% neutral buffered formalin for 24 h were dehydrated with ethanol for 30 min and embedded in paraffin (Paraplast Plus; Diapath, Belgamo, Italy). Paraffin sections of the embedded sample were cut using a microtome (Reichert-Jung 2050-supercut, Leica, Wetzlar, Germany) into serial 5 µm sections and stained with Mayer’s hematoxylin and eosin (H&E) and observed under a light microscope (Eclipse E-400, Nikon, Tokyo, Japan). Images were acquired by a Nikon digital light system (Nikon, Tokyo, Japan).

## Figures and Tables

**Figure 1 pathogens-10-01232-f001:**
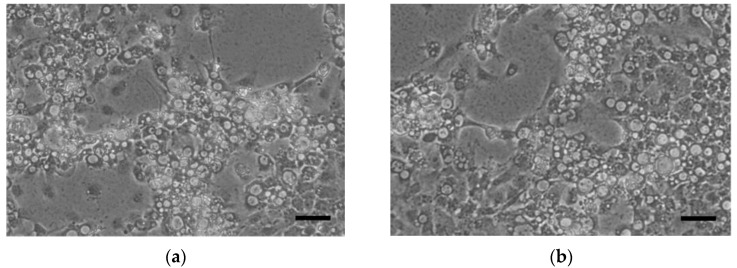
Cytopathic effect (CPE) on E-11 cells inoculated with tissue homogenate of shellfish positive for reassortant virus. After 5 days of the second round in blind passage, CPE was observed in E-11 inoculated with the domestic granular ark that contained the KS/RGNNV type (KG1-reKS/RG) (**a**) and the Manila clam imported from China that contained the RG/KSNNV type (CM1-reRG/KS) (**b**). Scale bar = 50 µm.

**Figure 2 pathogens-10-01232-f002:**
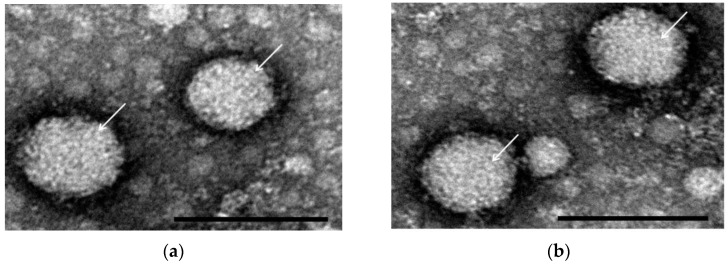
Electron micrograph of negatively stained virions of reassortant viruses. After negative staining with 1% uranyl acetate, virus particles (arrows) of KG1-reKS/RG (**a**) and CM1-reRG/KS (**b**) strains were observed under transmission electron microscopy (TEM) at a magnification of 200,000×. Scale bar = 50 nm.

**Figure 3 pathogens-10-01232-f003:**
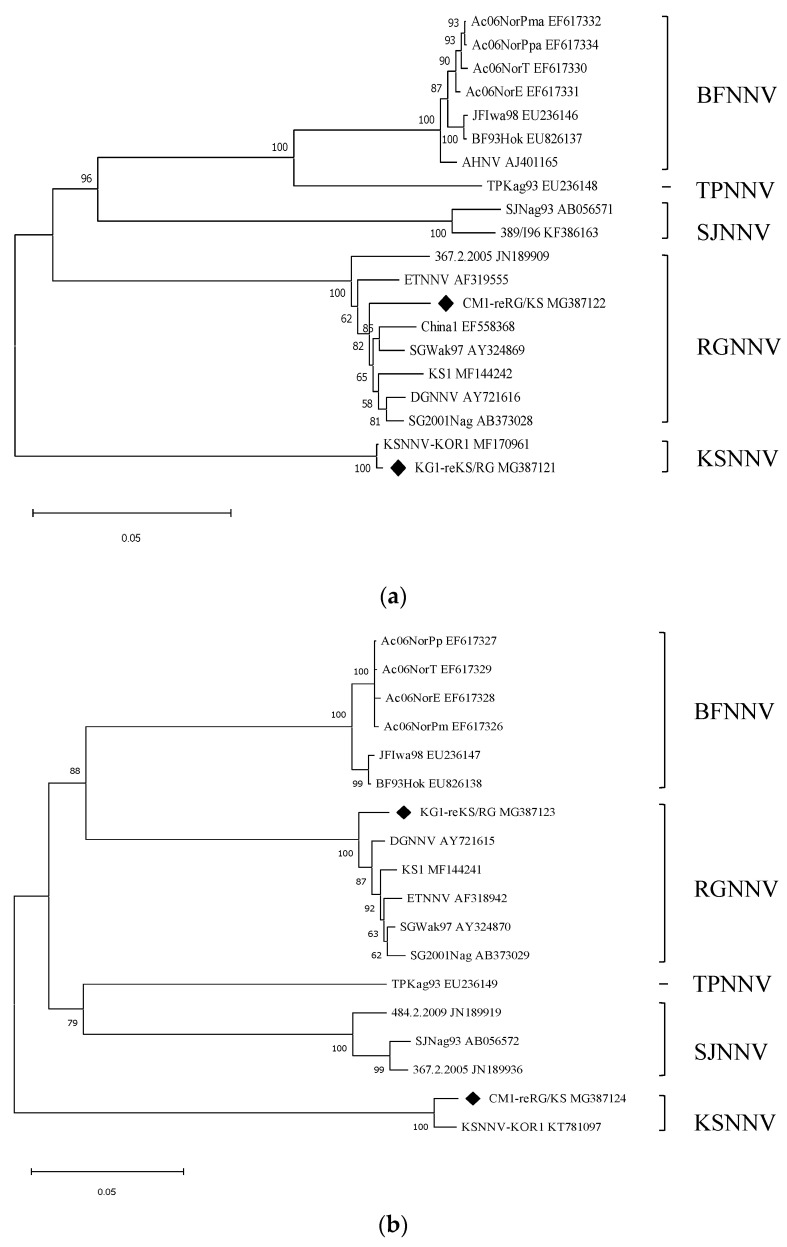
Phylogenetic tree based on the complete sequences of RNA1 and RNA2 segments of reassortant viruses identified in the present study. Phylogenetic relationship of RNA1 (**a**) and RNA2 (**b**) segments of reassortant viruses (♦) with reported betanodaviruses belonging to five types. Bootstrap values were obtained from 1000 replicates. The scale bar represents nucleotide substitution per site of 0.05. The GenBank accession number for the genomic sequence of each reported virus is indicated in parentheses.

**Figure 4 pathogens-10-01232-f004:**
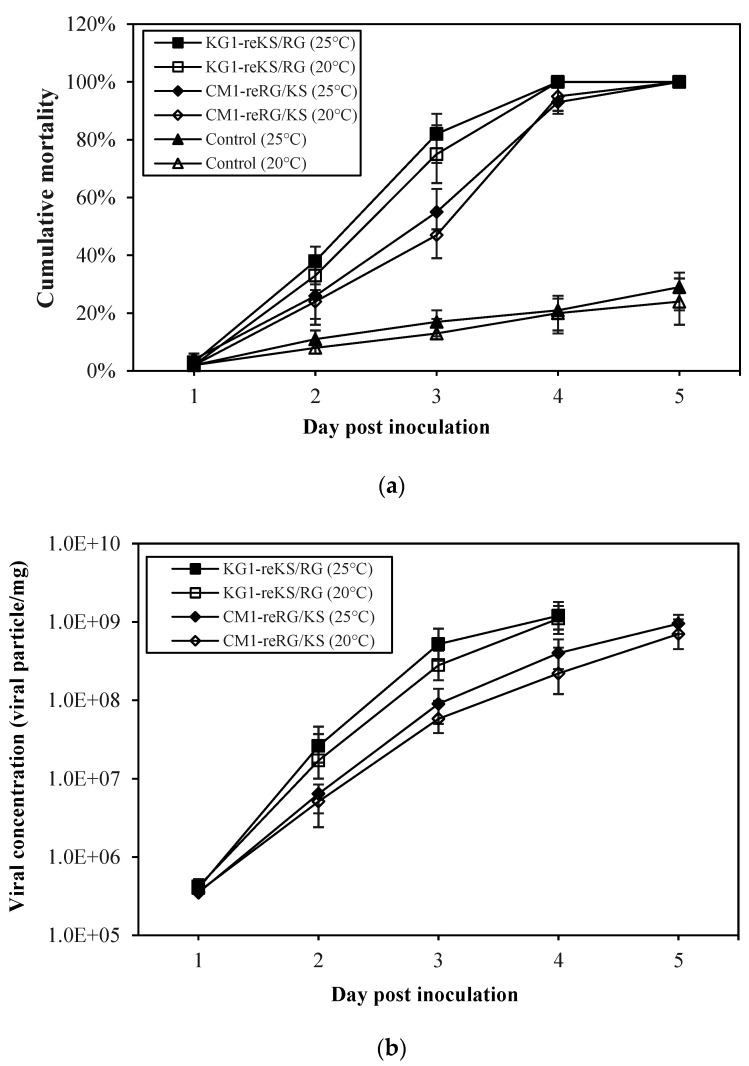
Cumulative mortality and viral growth kinetics of larval sevenband grouper bath challenged with two reassortant viruses at 25 °C and 20 °C. Sevenband grouper larvae seven days post-hatch (dpi) were exposed to KG1-reKS/RG and CM1-reRG/KS (10^7^ viral particles/mL) for four h in 2 L water by bath challenge. After inoculation, the cumulative mortality (**a**) and kinetics of viral growth in the whole larval bodies (**b**) were analyzed. Three individual sets of experiments were performed.

**Figure 5 pathogens-10-01232-f005:**
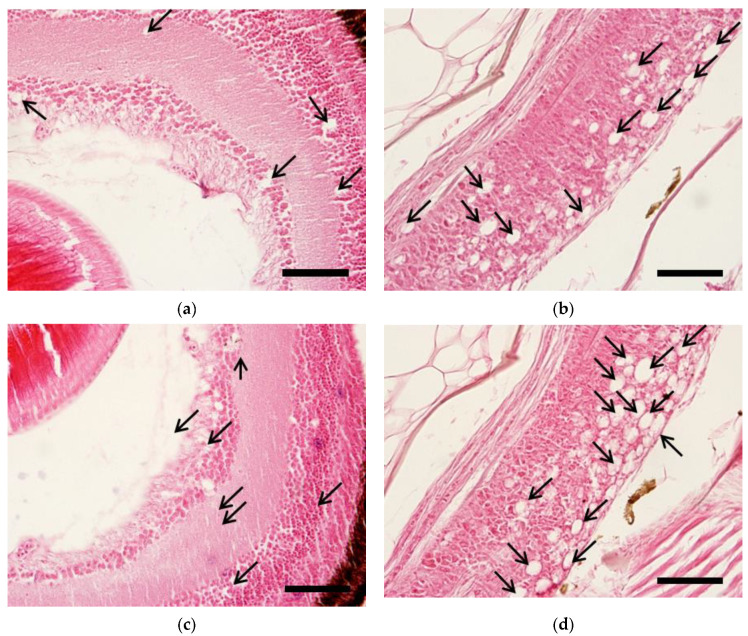
Vacuolation in the retina and brain tissue of moribund larval sevenband grouper infected with KG1-reRG/KS and CM1-reKS/RG three days post-inoculation (dpi). Severe vacuoles (arrows) in KG1-reKS/RG-infected retina (**a**) and brain (**b**) and CM1-reRG/KS-infected retina (**c**) and brain (**d**) of grouper larvae in both groups at 25 and 20 °C. Scale bar = 10 µm.

**Table 1 pathogens-10-01232-t001:** Identification of reassortment between betanodavirus types in domestic and imported shellfish and domestic finfish.

Samples	No. of Single Betanodavirus Positive ^1^	No. of Homologous Type ^2^	No. of Reassortant Virus ^3^		
			RG/KSNNV	KS/RGNNV	SJ/RGNNV
Domestic shellfish	306	302	1	1	2
Pacific oyster *Crassostrea gigas*	179	178	1	0	0
Mussel *Mytilus edulis*	49	49	0	0	0
Manila clam *Ruditapes philippinarum*	29	29	0	0	0
Common orient clam *Meretrix meretrix*	28	26	0	0	2
Granular ark *Tegillarca granosa*	21	20	0	1	0
Domestic finfish	35	35	0	0	0
Olive flounder *Paralichthys olivaceus*	23	23	0	0	0
Rock bream *Oplegnathus fasciatus*	3	3	0	0	0
Red sea bream *Pagrus major*	2	2	0	0	0
Rock fish *Sebastes schlegeli*	2	2	0	0	0
Stone flounder *Platichthys stellatus*	5	5	0	0	0
Imported shellfish from China	48	47	1	0	0
Manila clam *Ruditapes philippinarum*	14	13	1	0	0
Common orient clam *Meretrix meretrix*	13	13	0	0	0
Adams venus clam *Mercenaria mercenaria*	13	13	0	0	0
Venus clam *Mercenaria stimpsoni*	3	3	0	0	0
Wrinkled venus clam *Callista brevisiphonata*	3	3	0	0	0
Scallop *Saxidomus purpurata*	1	1	0	0	0
Chinese cyclina *Cyclina sinensis*	1	1	0	0	0
Imported shellfish from Japan	31	31	0	0	0
Manila clam *Ruditapes philippinarum*	5	5	0	0	0
Venus clam *Mercenaria stimpsoni*	5	5	0	0	0
Granular ark *Tegillarca granosa*	9	9	0	0	0
Scallop *Patinopecten yessoensis*	12	12	0	0	0
Total	420	415	2	1	2

^1^ Samples containg only a single type of betanodavirus in R2-DMT RT-PCR [10,11]; ^2^ Samples containing the same type of betanodavirus in the results between R1-DMT RT-PCR and R2-DMT RT-PCR; ^3^ Samples containing a betanodavirus showing an RNA1-RNA2 type mismatch in R1-DMT RT-PCR and R2-DMT RT-PCR.

**Table 2 pathogens-10-01232-t002:** Comparison of gene length of reassortant viruses with their corresponding betanodaviruses reported previously.

Isolates	Type	RNA1				RNA2				RNA3
		Total	RdRp	5′ UTR	3′ UTR	Total	CP	5′ UTR	3′ UTR	Protein B2
KG1-reKS/RG	KS/RGNNV	3123	2946	97	77	1434	1014	26	391	375
CM1-reRG/KS	RG/KSNNV	3105	2946	78	78	1444	1020	30	391	375
KSNNV-KOR1	KSNNV	3123	2946	97	77	1444	1020	30	391	375
SGWak97	RGNNV	3105	2946	78	78	1434	1014	26	391	375

**Table 3 pathogens-10-01232-t003:** Sequence homology of RNA1 and RNA2 segments of reassortant viruses compared with other types.

Segment	Part	Reassortant Strains	Homology to Corresponding Sequence (%)				
			KSNNV-KOR1 (KSNNV)	SGWak97 (RGNNV)	JFIwa98 (BGNNV)	SJNag93 (SJNNV)	TPKag93 (TPNNV)
RNA1	Total	KG1-reKS/RG	99.8 ^1^	82.7	82.2	81.2	82.5
		CM1-reRG/KS	83.2	97.2	83.1	82.2	82.8
	RdRp	KG1-reKS/RG	99.6 (99.8) ^2^	87.3 (88.6)	86.5 (87.3)	86.2 (87.2)	86.0 (87.1)
		CM1-reRG/KS	87.5 (89.6)	98.0 (98.7)	88.2 (89.6)	87.6 (88.9)	88.6 (89.5)
	RNA3	KG1-reKS/RG	99.5	81.0	80.6	80.9	81.5
		CM1-reRG/KS	82.8	94.1	83.4	81.6	84.5
	Protein B2	KG1-reKS/RG	99.6 (99.7)	83.8 (84.5)	83.3 (84.8)	83.3 (84.7)	82.5 (83.8)
		CM1-reRG/KS	85.5 (87.0)	95.2 (96.4)	86.8 (87.8)	83.3 (85.1)	87.3 (89.1)
	5′ UTR	KG1-reKS/RG	100.0	70.1	72.4	71.1	82.7
		CM1-reRG/KS	87.5	98.0	88.2	87.6	88.6
	3′ UTR	KG1-reKS/RG	98.7	84.6	79.0	84.6	84.6
		CM1-reRG/KS	84.6	96.2	79.7	82.3	82.3
RNA2	Total	KG1-reKS/RG	76.8	97.4	83.0	79.6	80.3
		CM1-reRG/KS	98.5	76.6	78.2	77.4	78.0
	CP	KG1-reKS/RG	76.0 (77.4)	99.7 (99.8)	86.4 (87.6)	81.5 (82.5)	81.5 (82.8)
		CM1-reRG/KS	98.8 (99.1)	76.3 (77.5)	78.9 (80.2)	77.5 (78.8)	79.2 (80.4)
	5′ UTR	KG1-reKS/RG	70.0	100.0	88.5	85.2	77.8
		CM1-reRG/KS	100.0	70.0	63.3	70.0	70.0
	3′ UTR	KG1-reKS/RG	78.3	98.0	83.2	80.2	81.4
		CM1-reRG/KS	98.7	77.7	82.2	80.2	79.8

^1^ Homology with corresponding nucleotide sequences; ^2^ Homology with deduced amino acid sequences.

**Table 4 pathogens-10-01232-t004:** Primers used in this study.

Primer	Sequence (5′ to 3′)	Object	Reference
c1VNNR	CCGTCTAATGCGACAGACATC	1st & 2nd round R1-DSN-2 RT-PCR	This study
c1VNNF1	GCGTTCCAAAAGAAAGAAGCATAC	1st round R1-DSN-2 RT-PCR	This study
c1VNNF2	GTTCCGTGGTACATGCCAAC	2nd round R1-DSN-2 RT-PCR R1-DMT RT-PCR	This study
s1VNNKSR	AAGCTCGTCAGCCACGATG	R1-DMT RT-PCR (for KSNNV)	This study
s1VNNRGR	TCTCATTAGCCAATAAAGTTGTTA	R1-DMT RT-PCR (for RGNNV)	This study
s1VNNBFR	CAGTATCAGTGAGGAGGGTGTC	R1-DMT RT-PCR (for BFNNV)	This study
s1VNNSJR	CATCATCCATGCCAGCTTG	R1-DMT RT-PCR (for SJNNV)	This study
s1VNNTPR	CAGCCAATATCCTCAATTTCG	R1-DMT RT-PCR (for TPNNV)	This study
c2VNNR	TGGTCATCAACGATACGC	1st & 2nd round R2-DSN-2 RT-PCR	[10]
c2VNNF1	GCTTCCTGCCTGATCCAAC	1st round R2-DSN-2 RT-PCR	[10]
c2VNNF2	TGCCAAATGGTGGGAAAG	2nd round R2-DSN-2 RT-PCR qRT-PCR	[10]
q2VNNR	TTGTTGCCGACACACAGG	qRT-PCR	[10]
s2VNNKSR	CGCTTCTGCGTTGTTTGG	R2-DMT RT-PCR (for KSNNV)	[11]
s2VNNRGR	TTGAAGTTGTCCCAGATGC	R2-DMT RT-PCR (for RGNNV)	[10]
s2VNNBFR	GGTAGAGCCAAGAAGTATTGATTTG	R2-DMT RT-PCR (for BFNNV)	[10]
s2VNNSJR	GGCAACGGTTTGTCAGTGAC	R2-DMT RT-PCR (for SJNNV)	[10]
s2VNNTPR	AAACCCAGAAGTGGCAGTTG	R2-DMT RT-PCR (for TPNNV)	[10]
R1KS-5 GSPR1	GGGTCACCAAATGGGGCCATCC	5′ RACE for KSNNV RNA1	[17]
R1KS-5 GSPR2	CACACAGAGCGGTGATGCCTGTGATGTC		[17]
R1KS-3 GSPF1	GCGCAACAGCACTGCACGAACTGTCC	3′ RACE for KSNNV RNA1	[17]
R1KS-3 GSPF2	GAATGCCATCGTGGCTGACGAGCTTGG		[17]
R2KS-5 GSPR1	GTTGCCAACGAGGATGGTCCGAAACG	5′ RACE for KSNNV RNA2	[17]
R2KS-5 GSPR2	CCGACCTGGAGACGTCAGTCGCTGCTC		[17]
R2KS-3 GSPF1	CATGTGGAGAAGGCCGCAGGAGATGC	3′ RACE for KSNNV RNA2	[17]
R2KS-3 GSPF2	CAGCCGCGGCAGATACTGCTTCC		[17]
R1RG-5 GSPR1	TCGCCGAAGGGCGCAATACCG	5′ RACE for RGNNV RNA1	This study
R1RG-5 GSPR2	CCACGAGGTCGGGAGCACACATACATGG		This study
R1RG-3 GSPF1	TGGTGCCGGGCTTATCAGAGGAATTG	3′ RACE for RGNNV RNA1	This study
R1RG-3 GSPF2	GACTGGAATGACGTTGTAGCCAACGAG		This study
R2RG-5 GSPR1	CGTGTTTGCGGGGCACATTGG	5′ RACE for RGNNV RNA2	This study
R2RG-5 GSPR2	GGCAGGAGGTCGGGGACGATGGTTG		This study
R2RG-3 GSPF1	CAGCCCCGTCAAATCCTGCTGCCTGT	3′ RACE for RGNNV RNA2	This study
R2RG-3 GSPF2	CCGGTTCCCTAGTGCGTATCGTTGA		This study

## Data Availability

Publicly available datasets were analyzed in this study. All nucleotide sequences of reassortant viruses in this study can be found in the National Center for Biotechnology Information (NCBI) GenBank (Access. No. MG366159, MG366160, KX601151, KX601152, MG387121, MG387122, MG387123, and MG387124).

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
