# Peer review of "Emergence of Reassortment between a New and Reported Types of Betanodavirus in Shellfish"

_pathogens, 2021, doi:10.3390/pathogens10101232_

Round 1

Reviewer 1 Report

I am not sure if genotype is a suitable word in this case, but I also do not know any other approriate word.

Author Response

Response: Thank you for your comment. We dedicated our best efforts to improve the quality of the manuscript under your suggestions. Considering your comment, we reviewed the suitable word regarding the reassortant of betanovirus from several studies as well as the viral classification in the International Committee on Taxonomy of Viruses (ICTV). Especially, for the genetically reassortant strain of betanodavirus, each RNA 1 or RNA 2 was represented as “-type or - genotype” based on the phylogeny or sequence homology in several studies. Thus, to convey the exact meaning of the genetic reassortant in the manuscript, we also would like to suggest the suitable word as ‘genotype” coincidence with ICTV and other studies. 

Reviewer 2 Report

The work entitled “Emergence of reassortant between a new and reported types of betanodavirus in shellfish” represents a study on a new reassortant betanodavirus detected in Asia. After a first introduction on the pathogen, the authors describe the biomolecular characterization of the pathogen, its attempted isolation on cell monolayer and an experimental infection study on E. septemfasciatus fry. In my opinion this manuscript reports a relevant and rigorous study supported by an appropriate experimental design. I recommend publication of the manuscript, with only a few minor corrections. Please find more specific comments below.

Introduction

Line 33 please delete] before RNA and add of before 3.1 kb

Line 33 to 34 please include in a square bracket the following sentence (encoding RNA-dependent RNA polymerase (RdRp), and protein B2 in subgenomic RNA3)

Line 34 please correct the typing error: ))]

Line 35 please replace traditionally with Generally

Results

Line 65 to 67 please rephrase the sentence

Line 84 please the species of the five shellfish

Table 1 please add the scientific name of sampled species

Table 1. could you provide the exact provenience of the sampled fishes and molluscs?

Figure 1 please add the Scale bar to (a) and (b)

Discussion

Line 173 please include in bracket KSNNV

Materials and methods

Line 252-257 which primer did you used to perform the analysis described in those sentences

257-258 in my opinion the sentence has to be reported in materials and method

Line 263 to 264 please add the scientific names of shellfish and fishes

Line 341 please correct the typing error “invtrogen”

Line 347 please replace Escherichia coli with E. coli

Line 358 to 360 indicate which sequences were used, specifying provenance and infected fish species

Line 368 to 369 indicate the complete name of E 11 cell monolayers, the species of animal to which it belongs and the commercial brand.

Author Response

 Thanks for your comment. We dedicated our best efforts to improve the quality of the manuscript under your suggestions. In accordance with the suggestions, we have revised the ‘Materials and Methods, Result and discussion” sections, as well as minor details throughout the manuscript. Especially, we have added detailed sample information including the scientific name used in this study.

Introduction

Line 33 please delete] before RNA and add of before 3.1 kb

Response: We agree with your comment. As per your comment, we have revised the sentence (L 33).

Line 33 to 34 please include in a square bracket the following sentence (encoding RNA-dependent RNA polymerase (RdRp), and protein B2 in subgenomic RNA3)

 Response: We agree with your comment. As per your comment, we have added the sentence suggested by you (L 33-34).

Line 34 please correct the typing error: ))]

Response: We agree with your comment. As per your comment, we have revised the error (L34).

Line 35 please replace traditionally with Generally

Response: We agree with your comment. As per your comment, we have revised the word (L 35). 

Results

 Line 65 to 67 please rephrase the sentence

Response: We agree with your comment. As per your comment, we have revised the sentence more clearly (L71-72).

Line 84 please the species of the five shellfish

Response: We agree with your comment. As per your comment, we have added the species of the five shellfish (L 90-91).

Table 1 please add the scientific name of sampled species

Response: We agree with your comment. As per your comment, we have added the scientific names in Table 1.

Table 1. could you provide the exact provenience of the sampled fishes and molluscs?

Response: We agree with your comment. As per your comment, we have added the detailed sample information in 4.1 Samples (Material and Methods section) (L267-269 and L279-L288).

Figure 1 please add the Scale bar to (a) and (b)

 Response: We agree with your comment. As per your comment, we have added the scale bar in Figure 1.

Discussion

Line 173 please include in bracket KSNNV

 Response: We agree with your comment. As per your comment, we have included in bracket KSNNV (L187)

Materials and methods

Line 252-257 which primer did you used to perform the analysis described in those sentences 257-258 in my opinion the sentence has to be reported in materials and method

 Response: We agree with your comment. As per your comment, we have revised the detailed primer information to perfrom the analysis (L269-274).

Line 263 to 264 please add the scientific names of shellfish and fishes

Response: We agree with your comment. As per your comment, we have added the scientific names of shellfish and fishes (L280-288).

Line 341 please correct the typing error “invtrogen”

Response: We agree with your comment. As per your comment, we have revised the typing error (L365).

Line 347 please replace Escherichia coli with E. coli

Response: We agree with your comment. As per your comment, we have replaced the word (L374).

Line 358 to 360 indicate which sequences were used, specifying provenance and infected fish species

Response: We agree with your comment. As per your comment, we have added the sequence information used for analysis (L376-L383).

Line 368 to 369 indicate the complete name of E 11 cell monolayers, the species of animal to which it belongs and the commercial brand.

Response: We agree with your comment. As per your comment, we have added the complete name of E11 cells including the commercial brand (L398-399).